# Evaluation of Patient No-Shows in a Tertiary Hospital: Focusing on Modes of Appointment-Making and Type of Appointment

**DOI:** 10.3390/ijerph18063288

**Published:** 2021-03-22

**Authors:** Mi Young Suk, Bomgyeol Kim, Sang Gyu Lee, Chang Hoon You, Tae Hyun Kim

**Affiliations:** 1Severance Children’s Hospital, Yonsei University, 50-1 Yonsei-ro, Seodaemun-gu, Seoul 03722, Korea; A27771@yuhs.ac; 2Department of Public Health, Yonsei University, 50-1 Yonsei-ro, Seodaemun-gu, Seoul 03722, Korea; arasion12@gmail.com; 3Department of Preventive Medicine, College of Medicine, Yonsei University, 50-1 Yonsei-ro, Seodaemun-gu, Seoul 03722, Korea; LEEVAN@yuhs.ac; 4Seoul Health Foundation, 31 Maebongsan-ro, Mapo-gu, Seoul 03909, Korea; sihoon.you@gmail.com; 5Department of Healthcare Management, Graduate School of Public Health, Yonsei University, 50-1 Yonsei-ro, Seodaemun-gu, Seoul 03722, Korea

**Keywords:** no-show, health care resource planning, modes of appointment-making, types of appointment

## Abstract

No-show appointments waste resources and decrease the sustainability of care. This study is an attempt to evaluate patient no-shows based on modes of appointment-making and types of appointments. We collected hospital information system data and appointment data including characteristics of patients, service providers, and clinical visits over a three-month period (1 September 2018 to 30 November 2018), at a large tertiary hospital in Seoul, Korea. We used multivariate logistic regression analyses to identify the factors associated with no-shows (Model 1). We further assessed no-shows by including the interaction term (“modes of appointment-making” X “type of appointment”) (Model 2). Among 1,252,127 appointments, the no-show rate was 6.12%. Among the modes of appointment-making, follow-up and online/telephone appointment were associated with higher odds of no-show compared to walk-in. Appointments for treatment and surgery had higher odds ratios of no-show compared to consultations. Tests for the interaction between the modes of appointment-making and type of appointment showed that follow-up for examination and online/telephone appointments for treatment and surgery had much higher odds ratios of no-shows. Other significant factors of no-shows include age, type of insurance, time of visit, lead time (time between scheduling and the appointment), type of visits, doctor’s position, and major diagnosis. Our results suggest that future approaches for predicting and addressing no-show should also consider and analyze the impact of modes of appointment-making and type of appointment on the model of prediction.

## 1. Introduction

No-shows occur when a patient fails to attend a scheduled appointment with no prior notification to the hospital [1]. Terms identified as corresponding to “no-show” in existing literature included appointment breaking, nonattendance, dropping out, missed appointment, and appointment failures [2]. No-show appointments not only decrease the sustainability of care for individuals but also pose several challenges for health care systems, including wasted resources, longer wait times, and concomitant threats to future patient satisfaction [3,4]. It is a well-known fact that no-show decreases the service provider’s productivity and efficiency, increases healthcare costs, and limits the medical institution’s effective capacity [5]. Hence, from the perspectives of sustainable medical services and operational efficiencies, managing patients’ no-show behavior is essential [6].

The growing number of patients and limited service capacity have greatly increased the need for a hospital appointment system [7]. Although there are some differences based on size or type of medical institutions, on an average, 70–90% of all outpatient care cases are based on the appointment system in Korea [8]. From the perspective of patient experience management, the appointment system promotes the exchange of information between hospitals and patients by providing real-time information to allow patients to schedule appointments with their preferred doctors at a convenient time [6,9].

However, no-show rates range from 5% to 25% across different hospitals [10,11,12]. The literature suggests that socio-economic backgrounds, clinical environments, hospital characteristics, and types of appointment systems affect no-shows [10,12,13,14,15]. Past studies have found that no-shows occur in female, younger age, and Medical Aid I and II recipient patient groups [7,12,13,14,15]. Other factors shown to be associated with no-shows are region, day of the week and time of appointment, types of appointment (new or follow-up), and professional situation [12,16,17,18,19,20,21,22,23]. In addition, religious events and holidays, distance between the clinic and patients’ homes, remote area, lack of social support, and social deprivation had relevance to no-shows [24,25,26,27,28]. A study on the methodology of predicting no show utilized a hybrid probabilistic prediction framework based on the elastic net variable-selection methodology integrated with probabilistic Bayesian Belief Network [29].

In summary, several studies have been conducted to find important variables associated with no-shows. However, limited literature has examined whether the modes of appointment-making, types of appointment, and their interactions are associated with patient no-shows. There are several modes of appointment-making. To increase patient accessibility, many hospitals let patients schedule an appointment directly, either over the phone or online. Appointment types also matter in scheduling, provider efficiency, patient satisfaction, and maximizing patient revenues.

By identifying in which types of appointment patient no-shows occur most often and organizing appointments accordingly, hospitals can ensure more revenue, happy doctors, and satisfied patients. Furthermore, it has been reported that modes of appointment-making and occurrences of no-shows vary according to the types of appointment [12]. To our knowledge, no research in Korea has examined the factors of no-shows by considering the influence between modes of appointment-making and types of appointment.

This study aimed to identify the interactions between modes of appointment-making and types of appointment among factors related to no-shows. This study is significant in that it provides insight into important factors relative to no-shows and is the first study to exhibit interactions between modes of appointment-making and types of appointment.

This paper is organized as follows: In Section 2, Materials and Methods are summarized. Specifically, it addresses the subjects and data source, variables and measurement, and statistical analysis. Then, in Section 3, the results of this study are explained. In Section 4, demographic, appointment-related, and practice-related factors associated with no-shows are discussed, in addition to the limitations and strengths of this study. Finally, Section 5 concludes with the main findings of this study.

## 2. Materials and Methods

### 2.1. Subjects and Data Source

This study used data of patients who made an outpatient appointment at a tertiary hospital located in Seoul, Korea. The hospital is one of the largest general hospitals in Korea, operating 63 departments, and thus allowed a large number of and diverse patients to be included in the study sample. Data were collected from medical records and administration records of patients who had reserved outpatient care appointments from 1 September 2018 to 30 November 2018. In order to protect personal information, the patients’ personal identification number was used instead of the hospital registration number, which was also anonymized in the initial extraction stage. During this period, the total number of outpatients was 256,011, and the total number of appointments given was 1,252,127.

This study was reviewed by the Yonsei University Health System Institutional Review Board and was ruled exempt (IRB number: Y-2019-0097). Written informed consent from patients was waived as this was a secondary data analysis using de identified data.

### 2.2. Variables and Measurement

#### 2.2.1. Dependent Variable

We examined the outpatient nursing department records for each patient to check the no-show status. All patients’ appointments were categorized as either “no-show” or “show-up.” No-show was defined as a patient who did not attend outpatient care on the day of the appointment [10].

#### 2.2.2. Independent Variables

##### Demographic Characteristics

The demographic characteristics included were gender, age, region (based on the patient’s residence), and types of insurance. Patients were divided into seven groups based on their age (≤19, 20–29 years, 30–39 years, 40–49 years, 50–59 years, 60–69, and ≥70 years). We divided the regions of patient residence into three groups: Seoul, Incheon Gyeonggi, and other areas. This categorization was employed as the study hospital is located in Seoul, the largest city of Korea; the second group, Incheon and Gyeonggi, represents the two of the most populated regions in Korea following Seoul; lastly, the remaining 14 cities and do-provinces of Korea are categorized as other areas. We divided types of insurance into four groups: National Health Insurance, Medical Aid, Industrial Accident Compensation Insurance and Automobile Insurance, and International Insurance and Private Insurance.

##### Appointment Related Characteristics

Among the appointment related characteristics, we divided the modes of appointment-making into three groups: Follow-up, online or telephone, and walk-in. Follow-up appointments meant appointment for the next schedule after previous medical treatment in the hospital. Walk-in appointments indicate that a patient visited the hospital in person to make an appointment. We divided time of visit into six groups: before“ 9:00 a.m., 9:00 a.m.–11:00 a.m., 11:00 a.m.–13:00 p.m., 13:00 p.m.–15:00 p.m., 15:00 p.m.–17:00 p.m., and after 17:00 p.m. Days of a week were from Monday to Sunday, and Sundays included an examination reservation in addition to a treatment reservation. Lead time was the time between scheduling and the appointment and, in this study, was divided into less than 8 days, 8–14 days, 15–21 days, 22–28 days, 29–56 days, 57–84 days, and more than 85 days. Seven days represent one week; however, to avoid overlap among the periods, the weeks were indicated as days.

##### Practice-Related Characteristics

The practice-related characteristics included the types of visit, type of appointment, department, doctor’s positions, and patient’s major diagnosis. We divided type of visit into three groups: A (new patients at the study hospital), B (patients who had visited the clinical department before), and C (new patients at the clinical department, but those who had visited the study hospital before). There were three major appointment types: Consultation, examination, and treatment and surgery. We divided clinical departments into 10 groups: Internal medicine, surgery department, obstetrics and gynecology (OBGYN), pediatrics, ophthalmology, otolaryngology, dermatology, urology, neuropsychiatry, and others. The positions of the doctors were categorized into professional, fellow, and training positions. The categorization followed the progression of years and training required for a doctor to be specialized in a particular field in Korea as his or her seniority increases within the hospital. For instance, a doctor who is in a fellow position refers to a person who trains for 1–2 years at a department of his or her major after obtaining the license to become a resident doctor. Patients’ major diagnoses were classified according to the 22 major diagnoses based on the Korean standard classification of diseases-7 codes.

### 2.3. Statistical Analysis

A three-step analysis was performed. First, for all categorical variables, we used chi-square tests to calculate the distribution of patient characteristics according to no-show status. This test is commonly used to test association between two or more categorical variables. Second, multivariate logistic regression analysis was used to assess the factors associated with no-show (Model 1). Finally, the interaction term (“modes of appointment-making” x “types of appointment”) was included in Model 2. Multiple logistic regression was used for two main reasons. First, the no-show status, the dependent variable of this study, was a binary outcome (Show-up: 0; No-show: 1). Second, logistic regression is appropriate for handling relationships among outcome variables and independent variables. To control for multiple appointments by the same patient, we incorporated repeated measures by using the “repeated subject” option in the generalized estimating equation. Statistical significance was established at *p* < 0.05. SAS software (ver. 9.4; SAS Institute, Cary, NC, USA) was used for all calculations and analyses.

## 3. Results

### 3.1. Comparison of Characteristics between No-Shows and Show-Ups

Table 1 shows the comparison of characteristics between no-show and show-up. During the study period, the number of outpatient appointments was 1,252,127, and the no-show rate was 6.12%. Regarding demographic characteristics, the group of subjects who failed to show-up for their appointments comprised mostly men (6.4%), those ≤19 years (7.2%), patients residing in Seoul (6.5%), and recipients of Medical Aid (8.7%).

For appointment-related characteristics, regarding the modes of appointment-making, the no-show rate was higher for walk-in appointments (10.6%). In terms of the time of the visit, the no-show rate was higher for 9:00–11:00 a.m. category (8.7%) than at other times. Regarding the day of week, the no-show rate was higher on weekends (9.6%) than on the other days. Regarding, on lead time, the following groups had the highest no-show rates: Less than 8 days (8.1%), 8–14 days (6.9%), and 57–84 days (6.7%).

For practice-related characteristics, regarding types of visits, the no-show rate was the highest for re-visits (6.5%). Based on types of appointment, the following groups had the highest no-show rates: Examination (30.1%), treatment and surgery (16.6%), and consultation (4.5%). The department category others had the highest no-show rate of 8.1%, followed by dermatology (7.3%) and otolaryngology (6.7%). With regard to doctor’s position, the no-show rate was the highest for the fellow position (6.5%). Regarding patient’s diagnosis, diseases of the nervous system (G00-G99) had the highest no-show rate at 11.3%, followed by injury, poisoning, and certain other consequences of external causes (S00-T98) (10.2%), and diseases of the genitourinary system (N00-N99) (10.1%).

### 3.2. Factors Associated with No-Shows

Table 2 shows the results of the multivariate logistic regression (Model 1) identifying the factors associated with no-shows. Regarding the modes of appointment-making, follow-ups had a lower odds ratio of no-shows (OR = 0.86) than walk-in. The odds ratio for online/telephone-based appointments was not statistically significant. Patients who underwent planned examination, treatment, and surgery had a much higher odds ratio of no-shows than those with planned consultation (examination OR = 9.09; treatment and surgery OR = 4.51).

In the demographic characteristics of Model 1, for gender, the odds ratio was higher in males (OR = 1.05). Based on age, the odds ratio for other age groups was lower than for the group aged ≤19 years (30–39 OR = 0.91; 40–49 OR = 0.81; 50–59 OR = 0.80; 60–69 OR = 0.78; ≥70 OR = 0.92). The odds ratio for 20–29 groups was not statistically significant. For region, the odds ratio for patients residing in the Incheon·Gyeonggi area (OR = 1.07) was lower than that for patients residing in Seoul. Based on the type of insurance, the odds ratio for Medical Aid (OR = 1.29) and International Insurance and Private Insurance (OR = 1.45) were higher than those for National Health Insurance.

In terms of appointment related characteristics of Model 1, based on the time of visit, the odds ratio for other times was higher than that for 9:00–11:00 a.m. (11:00 a.m.–13:00 p.m. OR = 1.16; 13:00 p.m.–15:00 p.m. OR = 1.10; 15:00 p.m.–17:00 p.m. OR = 1.18; ≥17:00 p.m. OR = 1.33). Weekends (Saturday, Sunday) had a higher odds ratio than Monday (OR = 1.43), but the odds ratio for Tuesday was lower (OR = 0.96). Based on the lead time, the odds ratios for 8–14 days (OR = 0.88), 15–21 days (OR = 0.81), and 22–28 days (OR = 0.88) were lower than that for less than 8 days. However, the odds ratios for 29–56 days (OR = 1.04) and 57–84 days (OR = 1.05) were higher.

From practice-related characteristics, based on the type of visits, the odds ratio for C (new patients at the clinical department, but those who had visited the study hospital before) (OR = 0.30) was lower than that for A (new patients at the study hospital). Based on the department, the odds ratios for surgery, ophthalmology, otolaryngology, dermatology, urology, and neuropsychiatry (OR = 1.15 to 2.15) were higher than that for internal medicine, however the odds ratios for OBGYN and others (OBGYN OR = 0.94; others OR = 0.71) were lower. For the doctor’s position, the odds ratios for training position (OR=1.24) and fellow position (OR = 1.16) were higher than for professional position. From the diagnosis point of view, compared to neoplasms (C00-D48), diseases of the nervous system (G00-G99) (OR = 1.15), diseases of the ear and mastoid process (H00-H59) (OR = 1.12), diseases of the respiratory system (J00-J99) (OR = 1.11), diseases of the digestive system (K00-K93) (OR = 1.12), diseases of the genitourinary system (N00-N99) (OR = 1.07), pregnancy, childbirth, the puerperium, and certain conditions originating in the perinatal period (O00-P96) (OR = 1.11), and symptoms, signs, and abnormal clinical and laboratory findings, NEC (R00-R99) (OR = 1.23) had higher odds ratios of no-shows. While several diseases had lower odds ratios than neoplasms (C00-D48), no diagnosis (OR = 0.44) was the lowest.

### 3.3. The Interaction Term

The interaction term is shown in Table 2. The interaction term between the modes of appointment-making and types of appointment was positive and statistically significant. As shown in Figure 1, for a follow-up patient, examination had a higher odds ratio of no-shows (OR = 12.59); however, treatment and surgery had a lower odds ratio (OR = 0.39). Patients with online telephone, treatment, and surgery had the highest odds ratio of no-shows (OR = 11.89).

## 4. Discussion

Identifying factors based on outpatient no-shows represents an important potential win–win for patients and hospitals by improving the sustainability of care and reducing wastage of resources. Our study demonstrates that demographic, appointment-related, and practice-related characteristics, which are factors readily available in the HIS data, can be successfully used to evaluate no-shows. Our study examined various factors associated with no-shows, focusing on the modes of appointment-making and types of appointment. The no-show rate in the study, 6.12%, is within the reported range of 5–25% according to the literature [10,11,12]. However, these studies had different timing of examination, examination period, and analysis perspective. Thus, it is difficult to directly compare the results of these studies with those of this study.

Several factors found to be influential for no-shows based on the literature review were also confirmed to be important factors in this study. The summary of the comparison between the factors associated with no-shows in this study and those in existing studies is as follows. First, our study showed that the odds ratio of no-shows for follow-up appointments in Model 2 considering interaction term. This finding might be related to how their needs are not addressed properly. Follow-up appointments are often scheduled after a patient has received a treatment, but it is scheduled to fit the department’s availability rather than the patients. However, in cases where patients themselves with medical conditions made online, telephone, or in-person appointments, the no-show ratio was lower because the appointment compliance was carried out actively by the patients according to their situation. Hospitals may need to prioritize appointment and medical service management with utmost consideration because high no-show ratio of follow-up patients has a direct and detrimental effect on their health [30,31,32].

Further, our study identified various factors including gender, age, region, types of insurance, time of visit, day of week, lead time, types of visit, types of appointment, department, doctor position, and major diagnosis as potential factors associated with no-shows.

In examining types of insurance, our study showed that the odds ratios for no-show for Medical Aid and International Insurance and Private Insurance were higher than that for National Health Insurance. This is similar to previous studies that showed that low-income patients covered by public assistance programs had higher odds of no-show [12,33,34], and patients with International Insurance and Private Insurance had higher odds of no-shows [12]. Medical institutions of Korea are categorized into three tiers: (1) Tertiary hospitals that provide specialized medical services, (2) primary medical institutions for basic medical services, and (3) clinics that have general practitioners for providing outpatient services [33]. A medical referral form issued by a sub-tier institution is required for patients to visit a tertiary hospital [35]. It is not difficult to obtain a referral at most medical institutions [35]. Patients with National Health Insurance visiting tertiary hospitals must pay 60% out-of-pocket expenses, while those with Medical Aid I pay a fixed amount of KRW 2000 and Medical Aid II pay 15% out-of-pocket expenses [36]. Thus, because Medical Aid patients can easily access high-tier medical institutions by paying relatively lower out-of-pocket expenses, they prefer tertiary hospitals that provide specialized medical services [35]. This phenomenon causes problems as the no-show rate increases. Moreover, owing to having no penalties—either financial or concerning future reservations—for no-shows, most patients are not watchful of no-shows. A no-show problem by Medical Aid patients has been brought up consistently as a major operative challenge for hospitals, suggesting the necessity of Medical Aid patient management.

Odds ratios for no-show were higher during Friday and weekends (Saturdays and Sundays), a finding similar to previous results that no-shows were most frequent on Fridays and Saturdays [37,38,39]. This result can be explained with the existing study that found more likelihood of no-shows in Friday reservations due to various events. On the contrary, some studies identified no-show were more likely to occur for Monday reservations than for weekend reservations. According to a study by Kwon et al. (2015) [12], which was conducted in the same study hospital, the days and likelihood of no-shows differed according to the types of appointment. In the case of consultation appointment, no-show was most frequent on Mondays, while Saturdays experienced the highest number of no-shows for a test, treatment, and surgery appointment. In the case of the study hospital, during weekends, especially on Sundays, most reservations are for diagnostic examinations, such as Magnetic Resonance Imaging (MRI), Computed Tomography (CT), and Positron Emission Tomography (PET), other than a consultation. Accordingly, contradictory results between this study and the existing study can be explained by differences in the type of appointment according to the days of the week. Moreover, this may occur partially because such time slots (i.e., Sunday) are generally less used. Thus, further studies should be conducted considering different days of the week.

Increase in lead time, which is the time between the booking of the appointment to the actual day of the appointment, seems to raise the odds ratio of no-shows. The further the date of appointment, the more likely the patient is to not show up. It is possible that with increased lead time, patients are more likely to forget their appointment, have a scheduling conflict, or visit another hospital or physician’s office that gives them a closer appointment date [10,33]. According to Athenahealth, “an analysis of 4.2 million appointments scheduled in 2016 by 13,000 providers found that shorter appointment lead times can be critical to getting new patients in the door” [40]. Specifically, Athenahealth found that “on average, a new patient who waits more than a month for a first appointment is more than twice as likely to cancel and not reschedule as a new patient who is scheduled within a week.” [40].

Lower odds ratios for established patients compared to new patients also have important implications. Established patients may have a reason to be attached to a specific hospital. They may have experienced the warmth of the staff, care of the nurses, and insightfulness of the physicians. However, to a new patient, a hospital clinic may still be just a name on a page. This is not unique to healthcare; it is a concept well studied in behavioral economics called the endowment effect [41].

In terms of departments, odds ratios of no-shows were higher for surgery department, pediatrics, ophthalmology, otolaryngology, dermatology, urology, and neuropsychiatry. This implies that tertiary hospitals, in managing ambulatory care, should invest greater efforts to improve patient compliance.

Furthermore, this study also proved that other major diagnoses had a higher no-show rate compared to neoplasms (C00-D48) along with their higher no-show ratio of examination, and treatment, and surgery appointments. This result could be related to patients’ needs. Tertiary care requires patients to have a medical referral form issued by primary or secondary care facilities, and these documents are issued quite easily by most medical institutions. Therefore, easy access to top medical institutions has resulted in more options for patients, which then leads to them favoring tertiary hospitals that provide specialized medical services [34,35]. Inevitably, this creates a problem of higher no-show rates for subordinate medical institutions as patients flock to tertiary medical institutions [33]. Considering the long waiting period in tertiary hospitals, patients with mild medical conditions who do not require immediate care tend to make appointments at several hospitals simultaneously [33]. This finding is important as any intervention aimed at reducing no-shows should include the effects of the patient’s diagnosis.

Our variables of key interest were modes of appointment-making, types of appointments, and their interactions. The finding that both follow-up and online/telephone appointments had higher odds ratios compared to walk-in appointments may suggest that relatively easier modes of appointment-making increase the likelihood of no-shows. Among the types of appointments, treatment and surgery had a much higher odds ratio of no-show compared to consultation. Missed treatment or surgery disrupts schedules, and potentially leaves doctors and nurses with gaps during the workday. When examining the interaction between the modes of appointment-making and types of appointment, follow-up and examination appointment had a higher odds ratio of no-shows while treatment and surgery had a lower odds ratio.

Not every examination will lead to an operation or admission, but doctors could lose out on service time for other patients when patients miss potentially critical examination appointments. Hospitals faced with a patient who does not show up for an examination may lose revenue, but the greater financial impact could be the possible necessary service that never occurs.

Another interaction term indicates that for patients who made appointments for treatment or surgery online or by phone had much higher odds of no-show. One might assume that providing direct or easy scheduling will result in better patient access, and thus, a reduction in no-shows. However, we could not find results that indicate that direct or easy scheduling improves patient access. Rather, the results suggest that direct or telephone scheduling decreases patients’ frequency of access to care.

Our study suggests that it is important for hospital managers to manage no-show with a focus on two aspects. First, we recommend that hospitals design interventions to reduce no-shows of follow-up patients who visit for examination. In addition, it is helpful for hospitals to decrease no-shows of online or telephone appointment patients who visit for treatment and surgery. Second, hospital managers must treat the issue seriously and design effective interventions for managing no-shows. In addition to satisfying the basic requirements of fundamental medical services, hospitals can provide other supportive services, for example, more comfortable waiting rooms and personalized accompanying services for patients.

This study has several limitations. First, this study was based on data on outpatient reservations at a single tertiary hospital for three months. Thus, the research results cannot be generalized to all tertiary hospitals. Second, restrictions on data extraction may have led to exclusion of certain variables that may be a factor in no-shows. In addition, the analysis did not include medical expenses, history of no-shows, and reason for no-shows, which were found to be associated with no-shows in the previous study. Therefore, to overcome the limitations of existing studies and this study, future studies need to include multiple medical institutions, as well as variables such as past no-show history and reservation change history. Despite these limitations, our study is unique in that it considered the modes of appointment-making and types of appointments simultaneously to examine no-shows and established the need to address patients’ medical needs. In addition, the target hospital responsible for this study is one of the largest tertiary hospitals in South Korea, which made it possible to analyze multiple and large groups of patients’ medical cases.

## 5. Conclusions

No-shows contribute to reduced scheduling efficiency and lowered effectiveness of the medical services delivered [42]. Sustainability of care is essential to ensure maximum health benefits for patients [43]. There is a need to develop interventions that will improve clinic attendance among clients [43]. Our results suggest that future approaches for predicting and addressing no-shows should also consider and analyze the impact of the modes of appointment-making and types of appointment on the model of prediction. Still, because no-shows constitute a complex issue and are affected by cultural differences and varying hospital systems, it is desirable to consider a conservative approach when applying the intervention, considering the limited generalizability of the present findings.

## Figures and Tables

**Figure 1 ijerph-18-03288-f001:**
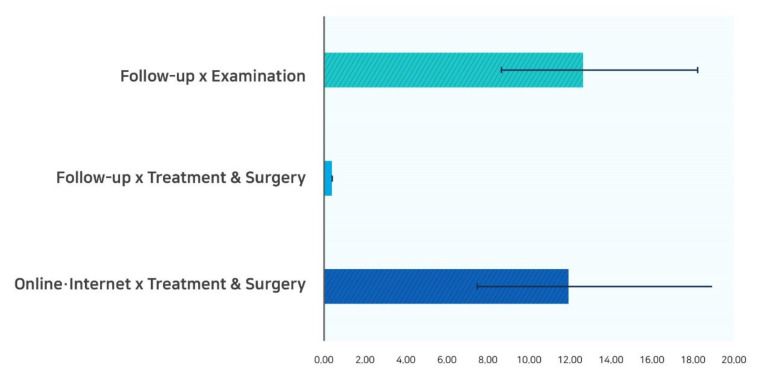
Multivariable logistic regression showing significant interaction between the modes of appointment-making and types of appointment.

**Table 1 ijerph-18-03288-t001:** Comparison between no-shows and show-ups.

Variables	Total		Show-Ups		No-Shows		*p*-Value
	N	%	N	%	N	%	
***Demographic characteristics***							
**Gender**							
Female	667,734	53.3	628,559	94.1	39,175	5.9	<0.0001
Male	584,393	46.7	546,895	93.6	37,498	6.4	
**Age**							
≤19	173,825	13.9	161,291	92.8	12,534	7.2	<0.0001
20–29	78,534	6.3	73,225	93.2	5309	6.8	
30–39	114,031	9.1	107,474	94.3	6557	5.8	
40–49	151,203	12.1	142,957	94.6	8246	5.5	
50–59	220,641	17.6	208,067	94.3	12,574	5.7	
60–69	249,549	19.9	235,529	94.4	14,020	5.6	
≥70	264,344	21.1	246,911	93.4	17,433	6.6	
**Region**							
Seoul	721,492	57.6	674,846	93.5	46,646	6.5	<0.0001
Incheon·Gyeonggi area	333,993	26.7	314,404	94.1	19,589	5.9	
Other areas	196,642	15.7	186,204	94.7	10,438	5.3	
**Types of insurance**							
National Health Insurance	1,166,189	93.1	1,096,750	94.1	69,439	5.9	<0.0001
Medical Aid	50,403	4.0	46,030	91.3	4373	8.7	
Industrial Accident Compensation Insurance and Automobile Insurance	10,929	0.9	10,164	93.0	765	7.0	
International Insurance and Private Insurance	24,606	2.0	22,510	91.5	2096	8.5	
***Appointment related characteristics***							
**Modes of appointment-making**							
Walk-in	59,187	4.7	52,922	89.4	6265	10.6	<0.0001
Follow-up	1,178,836	94.1	1,109,091	94.1	69,745	5.9	
Online/telephone	14,104	1.1	13,441	95.3	663	4.7	
**Time of visit**							
9:00–11:00	100,503	8.0	91,784	91.3	8719	8.7	<0.0001
<9:00	327,124	26.1	310,519	94.9	16,605	5.1	
11:00–13:00	269,994	21.6	253,514	93.9	16,480	6.1	
13:00–15:00	226,907	18.1	213,868	94.3	13,039	5.7	
15:00–17:00	236,243	18.9	221,798	93.9	14,445	6.1	
≥17:00	91,356	7.3	83,971	91.9	7385	8.1	
**Day of week**							
Monday	255,286	20.4	240,098	94.1	15,188	5.9	<0.0001
Tuesday	237,566	19.0	223,731	94.2	13,835	5.8	
Wednesday	226,738	18.1	213,466	94.2	13,272	5.8	
Thursday	266,325	21.3	250,448	94.0	15,877	6.0	
Friday	222,594	17.8	208,281	93.6	14,313	6.4	
Weekend (Saturday and Sunday)	43,618	3.5	39,430	90.4	4188	9.6	
**Lead time**							
Less than 8 days	231,134	18.5	212,432	91.9	18,702	8.1	<0.0001
8–14 days	158,800	12.7	147,811	93.1	10,989	6.9	
15–21 days	108,705	8.7	102,969	94.7	5736	5.3	
22–28 days	74,548	6.0	70,371	94.4	4177	5.6	
29–56 days	200,341	16.0	187,479	93.6	12,862	6.4	
57–84 days	102,606	8.2	95,705	93.3	6901	6.7	
More than 85 days	375,993	30.0	358,687	95.4	17,306	4.6	
***Practice-related characteristics***							
**Types of visit** *							
A	79,990	6.4	78,914	98.7	1076	1.3	<0.0001
B	1,015,836	81.1	949,339	93.5	66,497	6.5	
C	156,301	12.5	147,201	94.2	9100	5.8	
**Types of appointment**							
Consultation	1,098,863	87.8	1,049,756	95.5	49,107	4.5	<0.0001
Examination	15,831	1.3	11,059	69.9	4772	30.1	
Treatment and surgery	137,433	11.0	114,639	83.4	22,794	16.6	
**Department**							
Internal medicine	437,833	35.0	413,638	94.5	24,195	5.5	<0.0001
Surgery department	200,870	16.0	188,408	93.8	12,462	6.2	
OBGYN	64,137	5.1	61,235	95.5	2902	4.5	
Pediatrics	117,329	9.4	110,659	94.3	6670	5.7	
Ophthalmology	61,496	4.9	58,192	94.6	3304	5.4	
Otolaryngology	40,031	3.2	37,329	93.3	2702	6.7	
Dermatology	35,508	2.8	32,918	92.7	2590	7.3	
Urology	50,391	4.0	47,252	93.8	3139	6.2	
Neuropsychiatry	33,436	2.7	31,927	95.5	1509	4.5	
Others	211,096	16.9	193,896	91.9	17,200	8.1	
**Doctor’s position**							
Professional position	1,142,108	91.2	1,072,506	93.9	69,602	6.1	<0.0001
Training position	82,800	6.6	77,488	93.6	5312	6.4	
Fellow position	27,219	2.2	25,460	93.5	1759	6.5	
**Major diagnosis**							
A00-B99	12,821	1.0	11,866	92.6	955	7.4	<0.0001
C00-D48	159,483	12.7	146,806	92.1	12,677	7.9	
D50-D89	38,148	3.0	35,573	93.3	2575	6.7	
E00-E90	48,549	3.9	45,619	94.0	2930	6.0	
F00-F99	26,763	2.1	25,757	96.2	1006	3.8	
G00-G99	56,838	4.5	50,395	88.7	6443	11.3	
H00-H59	41,301	3.3	38,061	92.2	3240	7.8	
I00-I99	81,548	6.5	76,575	93.9	4973	6.1	
J00-J99	24,509	2.0	22,515	91.9	1994	8.1	
K00-K93	34,537	2.8	31,887	92.3	2650	7.7	
L00-L99	19,304	1.5	18,148	94.0	1156	6.0	
M00-M99	54,041	4.3	49,662	91.9	4379	8.1	
N00-N99	46,461	3.7	41,779	89.9	4682	10.1	
O00-P96	7025	0.6	6339	90.2	686	9.8	
Q00-Q99	17,635	1.4	16,263	92.2	1372	7.8	
R00-R99	56,723	4.5	51,220	90.3	5503	9.7	
S00-T98	19,544	1.6	17,546	89.8	1998	10.2	
U00-Z99	47,544	3.8	44,906	94.5	2638	5.5	
No diagnosis	459,353	36.7	444,537	96.8	14,816	3.2	

Note: * A: New patients at the study hospital; B: Patients who had visited the clinical department before; C: New patients at the clinical department, but those who had visited the study hospital before. Abbreviations: OBGYN, obstetrics and gynecology.

**Table 2 ijerph-18-03288-t002:** Multivariate logistic regression identifying the factors associated with no-shows.

Variables	Model 1				Model 2			
	OR	95% CI		*p*-Value	OR	95% CI		*p*-Value
***Demographic characteristics***								
**Gender**								
Female	ref							
Male	1.05	1.03	1.08	<.0001	1.05	1.03	1.08	<0.0001
**Age**								
≤19	ref							
20–29	1.05	0.99	1.11	0.1055	0.98	0.93	1.04	0.5295
30–39	0.91	0.86	0.97	0.0021	0.84	0.80	0.90	<0.0001
40–49	0.81	0.76	0.85	<.0001	0.74	0.70	0.79	<0.0001
50–59	0.80	0.76	0.84	<.0001	0.73	0.69	0.77	<0.0001
60–69	0.78	0.74	0.82	<.0001	0.71	0.67	0.75	<0.0001
≥70	0.92	0.87	0.97	0.0026	0.84	0.79	0.89	<0.0001
**Region**								
Seoul	ref							
Incheon·Gyeonggi area	0.97	0.94	0.99	0.0136	0.97	0.94	1.00	0.0359
Other areas	0.97	0.94	1.00	0.0701	0.98	0.95	1.01	0.1947
**Types of insurance**								
National Health Insurance	ref							
Medical Aid	1.29	1.21	1.37	<.0001	1.29	1.21	1.37	<0.0001
Industrial Accident Compensation Insurance and Automobile Insurance	0.96	0.82	1.13	0.6450	0.95	0.82	1.11	0.5459
International Insurance and Private Insurance	1.45	1.35	1.55	<.0001	1.35	1.26	1.45	<0.0001
***Appointment related characteristics***								
**Time of visit**								
9:00–11:00	ref							
<9:00	1.00	0.95	1.06	0.9182	0.98	0.93	1.04	0.5394
11:00–13:00	1.16	1.12	1.20	<.0001	1.15	1.12	1.19	<0.0001
13:00–15:00	1.10	1.06	1.14	<.0001	1.09	1.05	1.13	<0.0001
15:00–17:00	1.18	1.14	1.22	<.0001	1.17	1.13	1.21	<0.0001
≥17:00	1.33	1.27	1.39	<.0001	1.31	1.25	1.37	<0.0001
**Day of week**								
Monday	ref							
Tuesday	0.96	0.92	0.99	0.0121	0.96	0.93	0.99	0.0236
Wednesday	0.97	0.94	1.01	0.1394	0.97	0.94	1.01	0.1742
Thursday	0.98	0.95	1.02	0.3351	0.99	0.95	1.02	0.4626
Friday	1.03	0.99	1.07	0.0963	1.03	1.00	1.07	0.0773
Weekend (Saturday and Sunday)	1.43	1.35	1.53	<.0001	1.45	1.36	1.54	<0.0001
**Lead time**								
Less than 8 days	ref							
8–14 days	0.88	0.85	0.92	<.0001	0.88	0.85	0.92	<0.0001
15–21 days	0.81	0.78	0.85	<.0001	0.80	0.77	0.84	<0.0001
22–28 days	0.88	0.84	0.93	<.0001	0.88	0.83	0.92	<0.0001
29–56 days	1.04	1.00	1.09	0.0445	1.04	1.00	1.09	0.0470
57–84 days	1.05	1.00	1.10	0.0447	1.05	1.00	1.11	0.0411
More than 85 days	0.98	0.94	1.02	0.2422	0.96	0.92	1.00	0.0358
***Practice-related characteristics***								
**Types of visit** *								
A	ref							
B	1.01	0.98	1.04	0.4891	0.95	0.93	0.98	0.0019
C	0.30	0.28	0.32	<.0001	0.29	0.27	0.32	<0.0001
**Department**								
Internal medicine	ref							
Surgery department	1.46	1.42	1.51	<.0001	1.39	1.35	1.44	<0.0001
OBGYN	0.99	0.93	1.04	0.6083	0.94	0.89	0.99	0.0312
Pediatrics	1.14	1.07	1.21	<.0001	1.02	0.96	1.08	0.5791
Ophthalmology	1.21	1.14	1.29	<.0001	1.15	1.09	1.23	<0.0001
Otolaryngology	1.40	1.32	1.48	<.0001	1.35	1.28	1.43	<0.0001
Dermatology	2.21	2.08	2.36	<.0001	2.15	2.01	2.29	<0.0001
Urology	1.30	1.23	1.37	<.0001	1.27	1.20	1.34	<0.0001
Neuropsychiatry	1.59	1.45	1.74	<.0001	1.58	1.44	1.73	<0.0001
Others	0.67	0.64	0.71	<.0001	0.71	0.68	0.74	<0.0001
**Doctor’s position**								
Professional position	ref							
Training position	1.24	1.19	1.29	<.0001	1.26	1.21	1.31	<0.0001
Fellow position	1.16	1.09	1.24	<.0001	1.18	1.11	1.26	<0.0001
**Major diagnosis**								
C00-D48	ref							
A00-B99	0.96	0.89	1.03	0.2576	0.94	0.87	1.01	0.0912
D50- D89	0.95	0.91	1.00	0.0405	0.95	0.90	0.99	0.026
E00-E90	0.91	0.86	0.95	0.0001	0.88	0.84	0.93	<0.0001
F00-F99	0.40	0.36	0.45	<.0001	0.39	0.35	0.43	<0.0001
G00-G99	1.15	1.10	1.21	<.0001	1.13	1.08	1.18	<0.0001
H00-H59	1.14	1.08	1.21	<.0001	1.13	1.07	1.19	<0.0001
I00-I99	0.92	0.88	0.96	0.0002	0.89	0.86	0.93	<0.0001
J00-J99	1.11	1.05	1.18	0.0002	1.08	1.02	1.15	0.0057
K00-K93	1.12	1.06	1.18	<.0001	1.11	1.05	1.17	<0.0001
L00-L99	0.39	0.36	0.43	<.0001	0.39	0.35	0.43	<0.0001
M00-M99	0.93	0.88	0.98	0.0044	0.91	0.86	0.96	0.0002
N00-N99	1.07	1.02	1.13	0.0037	1.06	1.01	1.11	0.0263
O00-P96	1.11	1.01	1.22	0.0370	1.06	0.96	1.16	0.2732
Q00-Q99	0.92	0.86	0.98	0.0115	0.92	0.87	0.98	0.0143
R00-R99	1.23	1.18	1.28	<.0001	1.20	1.15	1.24	<0.0001
S00-T98	1.06	0.99	1.13	0.1003	1.04	0.97	1.11	0.2939
U00-Z99	0.88	0.85	0.92	<.0001	0.88	0.84	0.92	<0.0001
No diagnosis	0.44	0.43	0.46	<.0001	0.42	0.41	0.43	<0.0001
**Modes of appointment-making**								
Walk-in	ref				ref			
Follow-up	0.86	0.82	0.91	<.0001	1.36	1.24	1.49	<0.0001
Online/telephone	1.11	0.99	1.24	0.0619	1.36	1.18	1.57	<0.0001
**Types of appointment**								
Consultation	ref				ref			
Examination	9.09	8.66	9.54	<.0001	0.77	0.53	1.11	0.1610
Treatment and surgery	4.51	4.31	4.72	<.0001	9.38	8.47	10.39	<0.0001
**Modes of appointment-making** **× Types of appointment**								
Follow-up × Examination					12.59	8.67	18.27	<0.0001
Follow-up × Treatment and Surgery					0.39	0.35	0.44	<0.0001
Online/telephone × Treatment and Surgery					11.89	7.45	18.99	<0.0001

Note: * A: New patients at the study hospital; B: Patients who had visited the clinical department before; C: New patients at the clinical department, but those who had visited the study hospital before. Abbreviations: OR, odds ratio; CI, confidence interval; OBGYN, obstetrics and gynecology.

## Data Availability

The data presented in this study are available on request from the corresponding author.

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
