# Peer review of "Evaluation of Patient No-Shows in a Tertiary Hospital: Focusing on Modes of Appointment-Making and Type of Appointment"

_ijerph, 2021, doi:10.3390/ijerph18063288_

Round 1

Reviewer 1 Report

Dear author,

thank you very much for the opportunity to review your work.

I absolutely agree that investigation of couses for no-shows is essential for leveling hospital capacity in an efficent way. It also contributes to patient safety as no-shows may result in a loss of follow-up in critical health conditions.

However there are serious points that have to be adressed in you paper by rephrasing you objective and giving context to non-Korean audience who is neither familiar with geographic or socioeconomic details of Korean health care system (e.g type of health insurance).

Some remarks to be considered for resubmission:

#22: "predicts": this phrase needs to be revised. What do you want to state?

#54: "gender" "health insurance": please give details. otherwise rephrase this point

#63pp.: This section seems to sum up some results mixed with your hypothesis. However the introduction should only present the rational leading to your main hypothesis. The objective of your study is not clear at this point. Please end this section with the objective of your investigation. Please stress out the novelty of your approach comparing to previous studies.

#76: "catering" I am not sure whether catering is the appropriate word in this context.

#94: please give details for non-Korean readers why you made this geographic classification.

#96/97: please stick to one annotation e.g. "Insurance" vs. "insurance" for technichal terms

#104: as there is an international readership I would suggest to use a.m/p.m. Please also indicate which category would meet an appointment at 11.00 a.m. (9:00-11:00 or 11:00-13:00), there should be no overlap e.g. 9:00-10:59, 11:00-12:59 etc.

#107: again there are overlaps in used categories

#119: this may not be clear for non-Korean readership, what is the difference between professional position and general position?

Table 1/2: is there any intention to flip 9:00-11:00 and <9:00 in Table 1/2?

#216: It is unclear to me, how this retrospective cross-sectional analysis is able to predict anything. Clearly this is interesting in order to generate a hypothesis that can be tested prospectively. However it is also unclear whether no show rates may depend on season or other changes.

#241: please explain why this is a problem for non-Korean audience

#243: that´s a closed circlular argument, please rephrase

also some minor typos, suggest to review this paper after revision by a native English speaker.

Good luck

Author Response

Response letter

Thank you for the thoughtful and constructive feedback you provided regarding our manuscript. We have incorporated changes that reflect the detailed suggestions you have graciously provided. The whole paper has been reviewed and modified to remove any misspelling and grammatical errors. To facilitate your review of our revisions, the following is a point-by-point response to your questions and comments. Thank you once again for your consideration of our paper.

Response to Reviewer #1

Comments 1: #22: "predicts": this phrase needs to be revised. What do you want to state?

Response: We truly appreciate your comment. We have revised this sentence so that reviewers and readers can understand the intended meaning of this study (p.1, lines 21-24).

Comments 2: #54: "gender" "health insurance": please give details. otherwise rephrase this point

Response: We agree with your assessment. Accordingly, we have revised the sentence. Please do let us know if the revision is insufficient and we hope that the revised sentence meets your expectations and the standard of your publication (p.2, lines 55-57).

Comments 3: #63pp.: This section seems to sum up some results mixed with your hypothesis. However, the introduction should only present the rational leading to your main hypothesis. The objective of your study is not clear at this point. Please end this section with the objective of your investigation. Please stress out the novelty of your approach comparing to previous studies.

Response: Thank you for providing these insights. We agree with your assessment. The sentence has been modified to show our main hypothesis in the paragraph. In particular, we focused on better representing the objective of this study. Also, as the comment given by Reviewer 2, we have added a paragraph for the structure of this study at the end of the Introduction (p.2, lines 63-87).

Comments 4: #76: "catering" I am not sure whether catering is the appropriate word in this context.

Response: Thank you for pointing out that the expression can be improved. We appreciate it. As such, we have rewritten the sentence so that it can accurately reflect the original meaning we intended (p.3, lines 91-93).

Comments 5: #94: please give details for non-Korean readers why you made this geographic classification.

Response: We agree with you that the old sentence is difficult for the non-Korean audience to grasp. We thank you for pointing out the issue. As such, we have added a sentence that explains why the geographical categorization has been made between Seoul (where the hospital subject to the study is located), nearby areas (Incheon, Gyeonggi), and other provinces (p.3, lines 115-119).

Comments 6: #96/97: please stick to one annotation e.g. "Insurance" vs. "insurance" for technical terms

Response: We truly appreciate you letting us know about the cautions that must be paid to using technical terms. Based on your insightful advice, we have revised the sentence (p.3, lines 121).

Comments 7: #104: as there is an international readership I would suggest to use a.m/p.m. Please also indicate which category would meet an appointment at 11.00 a.m. (9:00-11:00 or 11:00-13:00), there should be no overlap e.g. 9:00-10:59, 11:00-12:59 etc.

Response: Thank you for pointing out the blind spot we have missed addressing. We will strive to make the paper more considerate and clearer. We have changed the sentence to use a.m./p.m. The categorization has also been modified so that there would be no overlap (p.3, lines 128-129).

Comments 8: #107: again there are overlaps in used categories

Response: Based on this thoughtful comment, we have modified the categorization so that there would be no overlap (p.3, lines 131-135).

Comments 9: #119: this may not be clear for non-Korean readership, what is the difference between professional position and general position?

Response: To make this sentence clearer, we have changed “general” to “fellow” positions. We have also added an explanation on the categorization and the meaning of “fellow position.” (p.3-4, lines 145-150).

Comments 10: Table 1/2: is there any intention to flip 9:00-11:00 and <9:00 in Table 1/2?

Response: Thank you for bringing up an insightful and interesting question. We have chosen the period between 9:00-11:00, which has the highest number of reservations, as the reference. To locate the reference period (9:00-11:00) at the top, we have changed the positions.

Comments 11: #216: It is unclear to me, how this retrospective cross-sectional analysis is able to predict anything. Clearly this is interesting in order to generate a hypothesis that can be tested prospectively. However it is also unclear whether no show rates may depend on season or other changes.

Response: You have raised an important question. We have also found that the existing sentence somewhat differs from what we meant to express. Please kindly understand that we have missed this point. As such, we revised the sentence, adding the part that, although the no-show rate is included in the range confirmed in existing literature, it is difficult to compare thereof directly with this study because each study differs (p. 9, lines 246-255).

Comments 12: #241: please explain why this is a problem for non-Korean audience.

Response: We very much agree that it is important to give detailed explanations for the non-Korean audience. Because we had to revise the sentence (#241) based on the given comment with Comment #243, detailed explanations on the old sentence could not be added. We hope that the revised sentence meets the standard of the Journal (p. 10, lines 284-299).

Comments 13: #243: that´s a closed circlular argument, please rephrase.

Response: Thank you for providing these insights. We have once again realized the importance of avoiding closed circular arguments. As such, we have conducted another literature review on the relevant part and revised the overall draft (p. 10, lines 284-299).

Reviewer 2 Report

This paper seeks to predict patient no-shows based on appointment-making modes and types of appointments. This study proposes a multiple logistic regression model to determine the variables in the analysis. The manuscript is well written and organized and I believe it should be considered for publication in this journal. Here are my suggestions:

i) The introductions are very brief and superficial. I suggest the discussion of important studies of the current state of the art:
"A comprehensive review on smart decision support systems for health care." IEEE Systems Journal 13.3 (2019): 3536-3545. DOI: 10.1109/JSYST.2018.2890121
"Predicting pediatric clinic no-shows: a decision analytic framework using elastic net and Bayesian belief network." Annals of Operations Research 263.1 (2018): 479-499. DOI: 10.1007/s10479-017-2489-0
ii) Include the main contributions of the research in the introduction, as well as a paragraph on the study organization.
iii) As it is a study involving humans, information about the acceptance by the hospital's research ethics committee must be included.
iv) Lines 124-127 the authors say "First, for all categorical variables, we used the chi-square tests to calculate the distribution of patient characteristics according to no-show. Second, multivariate logistic regression analysis was used to assess the factors associated with no-show." What is the justification for using these approaches, i.e., why did the authors not seek other statistical methods to infer about the data?
v) Lines 163-165. "Patients who underwent planned examination, treatment, and surgery had a much higher odds ratio of no-shows than those with planned consultation." What is the reason for this finding?
vi) Improve the Fig. 1 quality.
vii) Include the paper limitations in the conclusion.

Author Response

Response letter

We appreciate the time and effort you have dedicated to providing insightful feedback on ways to strengthen our paper. We have incorporated changes that reflect the detailed suggestions you have graciously provided. The whole paper has been reviewed and modified to remove any misspelling and grammatical errors. To facilitate your review of our revisions, the following is a point-by-point response to your questions and comments. Thank you once again for your consideration of our paper.

Response to Reviewer #2

Comments 1: The introductions are very brief and superficial. I suggest the discussion of important studies of the current state of the art:
"A comprehensive review on smart decision support systems for health care." IEEE Systems Journal 13.3 (2019): 3536-3545. DOI: 10.1109/JSYST.2018.2890121
"Predicting pediatric clinic no-shows: a decision analytic framework using elastic net and Bayesian belief network." Annals of Operations Research 263.1 (2018): 479-499. DOI: 10.1007/s10479-017-2489-0

Response: First, we truly appreciate you recommending important studies that can support our paper. We have thoroughly and sincerely reviewed the studies you have recommended. As such, in the Introduction, we have revised sentences to better represent the hypothesis of this study and its difference with existing studies. Thank you (p. 2, lines 59-62).

Comments 2: Include the main contributions of the research in the introduction, as well as a paragraph on the study organization.

Response: We agree and also think that it is important to express the main contributions of this study well in the Introduction. In order to better represent the main contributions of this study, we have supplemented thereof. In addition, we have added the organization of this study at the end of the Introduction (p. 2, lines 76-87).

Comments 3: As it is a study involving humans, information about the acceptance by the hospital's research ethics committee must be included.

Response: Thank you for raising an important question. We added the information on IRB (p.3, lines 100-102).

Comments 4: Lines 124-127 the authors say "First, for all categorical variables, we used the chi-square tests to calculate the distribution of patient characteristics according to no-show. Second, multivariate logistic regression analysis was used to assess the factors associated with no-show." What is the justification for using these approaches, i.e., why did the authors not seek other statistical methods to infer about the data?

Response: Thank you for raising an important question. We acknowledge that the statistical methods have certain limitations. As such, we have added the reasons why the statistical methods were chosen in this study (p. 4, lines 156-157; 160-163).

Comments 5: Lines 163-165. "Patients who underwent planned examination, treatment, and surgery had a much higher odds ratio of no-shows than those with planned consultation." What is the reason for this finding?

Response: We apologize that the existing statement on the study results is unclear. In the case of the sentence, it was written based on the analysis result of Model 1 in Table 2.

Comments 6: Improve the Fig. 1 quality.

Response: We have also been aware of the fact that the quality of Fig 1 will help the audience intuitively understand this study. Accordingly, we truly appreciate your comment. To improve the quality of Fig. 1, we used Adobe Illustrator to modify it.

Comments 7: Include the paper limitations in the conclusion.

Response: We have added the limitation of the study in the conclusion. Thank you (p. 12, lines 382-385).
